# Behavioral assessment of children and adolescents with Graves' disease: A prospective study

**Sherifa Ahmed Hamed**[1]*, **Fadia Ahmed Attiah**[1], **Samir Kamal Abdulhamid**[2], **Mohamed Fawzy**[1]

**1** Department of Neurology and Psychiatry, Assiut University Hospital, Assiut, Egypt, **2** Department of Internal Medicine, Assiut University Hospital, Assiut, Egypt

* hamed_sherifa@yahoo.com, hamedsherifa@aun.edu.eg

**Data Availability Statement:** Data cannot be shared publicly because they contain potentially sensitive information. Data are available from the ethics committee of Faculty of Medicine, Assiut, University (medicinegraduate@aun.edu.eg).

## Abstract

Previous studies have identified frequent comorbid neuropsychiatric disorders and conditions in adults with thyrotoxicosis. These studies are scarce or even lacking in pediatric population. This work aimed to study the behavior of children and adolescents with Graves' disease (GD). This study included 35 children with GD (boys = 15; girls = 25; mean age: 11.45±1.50yrs) and 40 healthy children (boys = 20; girls = 20; mean age: 12.54±1.62yrs). Behavior was assessed using Child Behavior Checklist (CBCL). Children with GD were assessed during periods of thyroid hormone elevation (active disease) and normalized thyroid hormones (with anti-thyroid drugs or ATDs). Compared to healthy children, patients during periods of thyroid hormone elevation (74.29%) and normalized thyroid hormones (31.43%) had higher frequencies of behavioral abnormalities and scorings of total CBCL scale (P = 0.01; P = 0.04, respectively) and its subscales' [Anxious/Depressed (P = 0.02; P = 0.04), Withdrawn/Depressed (P = 0.03; P = 0.04) and Somatic Complaints (P = 0.03; P = 0.127) and Social (P = 0.01; P = 0.225), Thought (P = 0.01; P = 0.128) and Attention (P = 0.01; P = 0.01) problems], indicating internalizing and externalizing problems. The majority of patients had at least two different behavioral problems. Marked improvement was found during period of normalized thyroid hormones (P = 0.001). Correlation analyses showed significant associations between total CBCL scoring and age at onset (P = 0.01; P = 0.001) and lower concentrations of thyroid stimulating hormone (TSH) (P = 0.001; P = 0.04) and higher concentrations of free thyroxine (fT4) (P = 0.01; P = 0.02), triiodothyronine (fT3) (P = 0.01; P = 0.03) and thyrotropin receptor antibodies (TRAbs) (P = 0.001; P = 0.01) during periods of thyroid hormone elevation and normalized thyroid hormones, respectively. Multiple linear regression analysis showed that "at presentation" lower concentrations of TSH (P = 0.001; P = 0.03) and higher concentrations of fT4 (P = 0.001, P = 0.01), fT3 (P = 0.01; P = 0.06) and TRAbs (P = 0.001; P = 0.001) were predictors of behavioral problems during periods of active disease and normalized thyroid hormones. We conclude that GD is associated with higher frequencies and severities of anxiety, depression and inattention during periods of thyroid hormone elevation as well as normalized thyroid hormones with ATDs. Therefore, early diagnosis and optimizing management are required to improve children's social life.

**Funding:** The authors received no specific funding for this work.

**Competing interests:** The authors have declared that no competing interests exist.

## Introduction

Graves' disease (GD) is the most common cause of childhood hyperthyroidism. It accounts for 10–15% of thyroid diseases in children less than 18 years old [1]. It may occur at any age during childhood and has its peak during adolescence [2]. GD is due to a complex interaction between genetics, environment, and immune system [3]. For many years, psychiatric assessment of adults with different states of hyperthyroidism [subclinical, overt or normal thyroid hormone levels' state with anti-thyroid drugs (ATDs) or thyroidectomy], is the focus of some studies and reviews. These studies documented the occurrence of more severe somatic manifestations beyond the integral manifestations of thyrotoxicosis and other comorbid neuropsychiatric conditions and disorders [4–6]. However, the incidence and prevalence of these neuropsychiatric comorbidities are not known. Reviews of literature documented that the frequencies of psychiatric symptoms with thyrotoxicosis vary widely from less than 10 to more than 70% [4–7]. Studies which assessed behavior in children with thyrotoxicosis are scarce. The frequently reported neurobehavioral manifestations in children with thyrotoxicosis include cognitive deterioration or poor scholastic achievement, hyperactivity, irritability or anxious dysphoria, and problems of attention [8, 9]. However, it is not clear whether the neuropsychiatric complications of thyrotoxicosis (whether in children or adults) follow a course parallel to the resolution of thyrotoxicosis or remain the same even after achievement of normal thyroid hormone levels with ATDs or thyroidectomy.

This study aimed to: (1) analyze behavior of children and adolescents with GD during periods of thyroid hormone elevation (active disease) and normalized thyroid hormones with ATDs. The Child Behavior Checklist (CBCL) was used to assess behavior, and (2) study the independent demographic, clinical and laboratory variables which were associated with behavioral problems during periods of active disease and normalized thyroid hormones after one year of treatment with ATDs.

## Methods

Thirty five children and adolescents (girls = 25; boys = 10) with confirmed diagnosis of GD were included in this prospective study. Their age ranged from 9 to 18 years. Patients were recruited and followed over a period of two years (2017–2019) from the Clinical Medicine multidisciplinary follow-up out-patient clinic of Assiut University Hospital, Assiut, Egypt. The diagnostic criteria for GD included the presence of clinical hyperthyroidism and reduced thyroid-stimulating hormone (TSH), and elevated free thyroxine (fT4) and triiodothyronine (fT3) blood concentrations, and high titers of thyrotropin receptor antibodies (TRAbs) [10]. According to thyroid hormone status, children with GD were divided into: **(1)** those with active GD (i.e. newly diagnosed and did not previously receive ATDs], and **(2)** those with normal thyroid hormone levels (i.e. maintained in normalized thyroid state by ATDs for one year). Patients were treated with carbimazole in a dose of 0.5–0.7 mg/kg/day which was subsequently titrated to maintain normal thyroid hormone levels based on the results of serum thyroid hormone testing during follow-ups [11]. Also this study included 40 healthy children and adolescents matched for sex (girls = 20; boys = 20), age (range: 9–18 yrs; mean: 12.54 ± 1.62 yrs), pubertal status and educational and socioeconomic states. They were recruited from healthy schoolmates. Excluded from the study were children with: **(1)** mental developmental delay (i.e. intelligence quotient or IQ below 85) [12, 13], **(2)** other medical, systemic or autoimmune diseases, **(3)** previous relapse(s), **(4)** previous history of psychiatric disorders or brain insults, and **(5)** history of regular intake of B-blockers or psychotropic medications during the period of the study.

All patients underwent medical, endocrinology and neurological histories and examinations. The following data were collected: age at presentation, age of onset, duration and manifestations of GD, dose and duration of treatment with ATD(s) and duration of normalized thyroid hormone levels' state. Anthropometric measurements (height and weight), body mass index (BMI), blood pressure and heart rate were also recorded.

## Neurobehavioral assessment

**Assessment of intelligence.** Intelligence was assessed using Wechsler Intelligence Scale for Children, Third Edition (WISC-III) [13]. The scale consists of 6 verbal subtests (verbal IQ) (similarities, digit span, vocabulary, arithmetic, comprehension, and information) and five performance subtests (performance IQ) (picture completion and arrangement, coding subtest, digit symbol, and block design).

## Behavioral assessment

Interviewing of parents and children were done by the psychiatrists (FAA and MF). Child Behavior Check List (CBCL/6-18) was used for screening of emotions and behavior [14, 15]. Rating for different emotional, behavioral and social problems were defined by children's parents as either not true (0), somewhat or sometimes true (1), or very true or often true (2). Problem Checklist of CBCL (demonstrated as 112 statements) are classified into 3 subscales: **(1)** The narrow-band syndrome scales have 8 items [Anxious/Depressed, Withdrawn/Depressed and Somatic Complaints; Social, Thought and Attention Problems; and Rule-Breaking and Aggressive Behaviors], **(2)** Broad-band internalizing [Anxious/Depressed, Withdrawn/Depressed and Somatic Complaints; and Social and Thought Problems] and externalizing [Attention Problems; Rule-Breaking and Aggressive Behavior] problems, and **(3)** Diagnostic and Statistical Manual of Mental Disorders (DSM) scales [anxiety, oppositional defiant and attention deficit disorders and affective, somatic and conduct problems]. CBCL/6-18 scores can be used as continuous variables or according to total scoring ranges. According to the total scoring ranges, children were classified as: **(1)** normal (score = 50–64), **(2)** borderline (score = 65–69) or **(3)** abnormal (score = 70–100).

## Laboratory testing

Blood was withdrawn at nearly 8 a.m. after overnight fasting for determination of serum concentrations of thyroid hormones (Immulite™ 2000 Third Generation, Diagnostic Products Corporation, Los Angeles, CA). Reference ranges for 'thyroid function tests are as follow: TSH: 0.4–4.0 mU/L, fT4: 10.0–26.0 pmol/L, and fT3: 3.5–5.5 pmol/L. Blood TRAbs concentrations were measured using 3rd generation TBII assay (TRAb3rd) (Elecsys, Roche Diagnostics GmbH, Penzberg, Germany). The reference cut-off value for positive concentration of TRAbs is 1.75 IU/L.

## Statistical analyses

Analysis was done using SPSS version 16.0 (Statistical Package for the Social Sciences Inc, Chicago III). The distribution of data was evaluated using the Kolmogorov-Smirnov test. Data were expressed as mean ± standard deviation (SD) as they had continuous distribution. Comparative statistics were done using Student's t- (two-tailed) and Chi-square tests. For each group (in the active and drug-corrected states), correlation analyses between total scoring of CBCL and variables (age at presentation and "at presentation" concentrations of TSH, fT4, fT3 and TRAbs) were done using Pearson's correlation coefficient. For each group, multiple linear

regression analysis was used to examine the relationship between total scoring of CBCL (dependent variable) and age at presentation and "at presentation" concentrations of TSH, fT4, fT3 and TRAbs (independent variables) in the active and drug-corrected states. Two models were examined; model A was adjusted for age at presentation and model B was additionally adjusted for concentrations of TSH, fT4, fT3 and TRAbs. The β value, 95% confidence intervals percentage (CI %) and significances were calculated. Significance was considered as probability less than 0.05 (P<0.05).

### Ethics statement

The ethics Committee of Faculty of Medicine of Assiut University, Assiut, Egypt approved the study protocol (ID# AUFM_243/2017). Parents/guardians gave their written inform consents to participate in the study.

### Results

Thirty five children with GD were included in this study (boys to girls ratio = 1: 2.5). They had mean age of 11.45 ± 1.50 yrs. All had normal neurological examinations and IQs. There were no differences between children 'with drug-corrected state and healthy children in thyroid laboratory markers (TSH, fT3, fT4 and TRAbs) (**Table 1**). Compared to healthy children, children with GD (whether in active or drug-corrected states) had higher total scoring of CBCL (P = 0.01; P = 0.04) particularly in Anxious/Depressed (P = 0.02; P = 0.04), Withdrawn/ Depressed (P = 0.03; P = 0.04) and Attention (P = 0.01; P = 0.01) Problems subscales'. No differences were identified between children with GD and healthy children in scores of Rule-Breaking and Aggressive Behaviors (**Table 2**). There were overlaps of different behavioral problems (i.e. at least 2 problems) in the majority of children with active and normalized thyroid hormone levels' states (particularly with active state). Higher frequencies of patients classified as at or above the borderline clinical stage for CBCL, were observed in children with active and normalized thyroid hormone levels' states particularly with active state (P = 0.001) (**Table 3**). In the two groups of patients, correlation analyses showed that total scoring of CBCL was correlated with age at onset (r = -0.355, P = 0.01; r = -0.506, P = 0.001), "at presentation" lower concentrations of TSH (r = -0.655, P = 0.001; r = 0.258, P = 0.04) and higher concentrations of fT4 (r = 0.432, P = 0.01; r = 0.306, P = 0.02), fT3 (r = 0.446, P = 0.01; r = 0.280, P = 0.03) and TRAbs (r = 0.663, P = 0.001; r = 0.320, P = 0.01). Multiple linear regression analysis showed that "at presentation" lower concentrations of TSH (P = 0.001; P = 0.03) and higher fT4 (P = 0.001, P = 0.01), fT3 (P = 0.01; P = 0.06) and TRAbs (P = 0.001; P = 0.001) were predictors of behavioral problems in children with active as well as normalized thyroid hormone levels' states (**Table 4**).

### Discussion

The emotional state and behavior of children and adolescents with GD were prospectively assessed during the periods of active disease and normalized thyroid hormone levels' with AEDs. Results of this study indicate that: **(1)** High frequencies of patients with active GD (74.29%) or normalized thyroid hormone levels' (31.43%) had high total scoring of CBCL particularly its Anxious/Depressed, Withdrawn/Depressed; and Somatic Complaints and Social, Thought and Attention Problems subscales, indicating internalizing and externalizing behavioral problems. **(2)** Marked improvement was observed after normalization of thyroid hormone levels with ATDs. **(3)** Anxiety and inattention were the prominent behavioral manifestations both during the active and normalized thyroid hormone levels' states. **(4)** Behavioral problems seem to be independently related to the severity of GD.

**Table 1. The demographic, clinical and laboratory characteristics of the studied groups.**

| Demographic, Clinical and Laboratory Characteristics | Patients with active GD | Patients in normalized thyroid hormone levels | Controls |
|---|---|---|---|
| | (n = 35) | (n = 35) | (n = 40) |
| **Demographic and Clinical Data** | | | |
| **Age at presentation;** yrs | 9–18 (11.45 ± 1.50) | - | 8–18 (12.54 ± 1.62) |
| P1-value | 0.323 | - | |
| **Gender** | | | |
| Girls | 25 (71.43%) | 25 (71.43%) | 20 (50%) |
| Boys | 10 (28.57%) | 10 (28.57%) | 20 (50%) |
| **Heart rate;** beats/minute | 90–130 (110.00 ± 3.00) | 80–100 (90.00 ± 5.00) | 75–105 (85.00 ± 5.00) |
| P1-value | **0.03** | 0.325 | |
| P2-value | | **0.01** | |
| **Systolic blood pressure;** mmHg | 100–130 (110.00 ± 5.00) | 90–110 (100.00 ± 0.00) | 95–110 (100.00 ± 5.00) |
| P1-value | 0.211 | 1.003 | |
| P2-value | | 0.212 | |
| **Diastolic blood pressure;** mmHg | 60–80 (70.00 ± 0.00) | 60–75 (65.00 ± 5.00) | 55–80 (65.00 ± 0.00) |
| P1-value | 0.456 | 1.002 | |
| P2-value | | 0.864 | |
| **Body Mass Index;** kg/m$^2$ | 16–25 (18.90 ± 4.30) | 18.5–26.9 (22.50 ± 2.30) | 18.5–28.9 (24.9 ± 2.30) |
| P1-value | 0.135 | 0.426 | |
| P2-value | | | |
| **Laboratory Data** | | | |
| **TSH;** μU/l | 0.005–0.22 (0.01 ± 0.006) | 0.37–3.55 (1.32 ± 0.45) | 0.62–4.42 (1.75 ± 0.86) |
| P1-value | **0.0001** | 0.345 | |
| P2-value | | **0.0001** | |
| **fT4;** pmol/l | 24.30–46.52(32.10 ± 2.68) | 7.25–16.33 (12.86 ± 1.36) | 7.54–14.96 (11.46 ± 1.45) |
| P1-value | | 0.520 | |
| P2-value | **0.0001** | **0.0001** | |
| **fT3;** pmol/l | 15.45–29.68 (19.45 ± 2.70) | 4.60–9.2 (5.50 ± 0.80) | 3.80–7.80 (5.26 ± 1.06) |
| P1-value | | 0.866 | |
| P2-value | **0.0001** | **0.0001** | |
| **TRAb;** IU/ml | 180.60–470.37 (265.74 ± 55.86) | 22.65–80.40 (42.36 ± 8.55) | 16.88–64.25 (46.45 ± 6.60) |
| P1-value | | 0.688 | |
| P2-value | **0.0001** | **0.0001** | |
| **Intelligence Quotient (IQ)** | | | |
| **Verbal IQ** | 80–130 (100.90 ± 10.28) | 80–140 (102.70 ± 15.60) | 90–145 (112.60 ± 18.55) |
| P1-value | 0.345 | 0.433 | |
| P2-value | | 0.864 | |
| **Performance IQ** | 70–135 (98.14 ± 12.33) | 70–145 (99.19 ± 13.58) | 86–132 (108.87 ± 15.60) |
| P1-value | 0.235 | 0.268 | |
| P2-value | | 0.824 | |
| **Full scale IQ** | 70–135 (101.18 ± 15.31) | 75–137 (100.06 ± 12.52) | 80–133 (105.90 ± 9.93) |
| P1-value | 0.632 | 0.636 | |
| P2-value | | 1.022 | |

Data are expressed as mean ± SD; number (%).

P1: significance versus controls; P2: significance versus those with active GD.

TSH: thyroid stimulating hormone; fT4: free thyroxine; fT3: free triiodothyronine; TRAbs, thyroid-stimulating hormone receptor antibodies.

**Table 2. Behavioral assessment categorization and scores for the studied groups.**

| Behavioral assessment | Patients with active GD | Patients in normalized thyroid hormone levels | Controls |
|---|---|---|---|
| | (n = 35) | (n = 35) | (n = 40) |
| **Anxious/Depressed** | 50–85 (74.37 ± 3.61) | 50–88 (68.03 ± 2.57) | 50–63 (55.42 ± 2.97) |
| P1-value | **0.02** | **0.04** | |
| P2-value | | 0.258 | |
| **Withdrawn/Depressed** | 50–89 (68.90 ± 3.33) | 50–94 (66.90 ± 3.58) | 50–76 (53.40 ± 4.43) |
| P1-value | **0.03** | **0.04** | |
| P2-value | | 0.655 | |
| **Somatic Complaints** | 50–80 (68.57 ± 4.16) | 50–84 (60.12 ± 5.81) | 50–79 (54.57 ± 5.45) |
| P1-value | **0.03** | 0.127 | |
| P2-value | | 0.264 | |
| **Social Problems** | 50–98 (70.97 ± 5.69) | 50–91 (60.72 ± 5.89) | 50–70 (52.13 ± 3.53) |
| P1-value | **0.01** | 0.225 | |
| P2-value | | 0.320 | |
| **Thought Problems** | 51–97 (72.10 ± 3.82) | 50–90 (62.33 ± 3.84) | 50–67 (52.73 ± 4.33) |
| P1-value | **0.01** | 0.128 | |
| P2-value | | 0.256 | |
| **Attention Problems** | 52–100 (73.03 ± 6.76) | 50–95 (72.44 ± 5.08) | 50–81 (52.12 ± 4.93) |
| P1-value | **0.01** | **0.01** | |
| P2-value | | 0.657 | |
| **Rule-Breaking Behavior** | 50–68 (56.00 ± 2.57) | 50–76 (55.74 ± 2.98) | 50–68 (54.12 ± 3.78) |
| P1-value | 0.355 | 0.332 | |
| P2-value | | 0.724 | |
| **Aggressive Behavior** | 50–100 (57.31 ± 8.70) | 50–79 (53.03 ± 6.43) | 50–70 (53.48 ± 3.69) |
| P1-value | 0.440 | 0.678 | |
| P2-value | | 0.545 | |
| **Total score** | 50–100 (70.53 ± 4.11) | 50–95 (64.85 ± 5.36) | 50–81 (51.33 ± 4.20) |
| P1-value | **0.01** | **0.04** | |
| P2-value | | 0.625 | |

Data were expressed as Mean ± SD.

Comparative statistics were done using Student's-t-test.

P1: patients versus controls; P2: patients with active GD versus those in normalized thyroid hormone levels.

In accordance, in adults with hyperthyroidism, authors observed incomplete normalization of psychopathological and neuropsychological changes after longer periods of normalization of thyroid hormone levels [4–6, 16, 17]. Bommer et al. [16] observed that adults with "subclinical" or "remitted" hyperthyroidism (n = 45) had the following: (a) more behavioral abnormalities which included reduced well-being with feelings of fear (43% versus 10% for controls), hostility, inability to concentrate, fearful-agitated syndrome (dominated mainly early at onset of the disease) and depression (dominated mainly after longer periods of normalized thyroid hormone levels), and (b) marked impairment of neuropsychological functioning particularly in relapse after 2.5 yrs (more than 25% versus 2% for controls). Paschke et al. [17] found significant reduction in anxiety, irritability, depression, exhaustion, ability to concentrate, extroversion and introversion manifestations and reduced well-being in adults with thyrotoxicosis after 1 to 2 months of achievement of normal thyroid hormone levels with ATDs or thyroidectomy. However, they observed that patients had never returned to baseline before the disease onset. Stern et al. [4] reported the following: (a) subclinical cognitive deficits (manifested as

**Table 3. Frequency of patients at or above the borderline clinical range for CBCL.**

| Behavioral assessment | Patients with active GD state | Patients with normalized thyroid hormone levels state | P-value |
|---|---|---|---|
| | (n = 35) | (n = 35) | |
| Anxious/Depressed | 26 (74.29%) | 15 (42.86%) | **0.01** |
| Withdrawn/Depressed | 11 (31.43%) | 8 (22.86%) | 0.08 |
| Somatic Complaints | 20 (57.14%) | 15 (42.86%) | **0.04** |
| Social Problems | 18 (51.43%) | 8 (22.86%) | **0.01** |
| Thought Problems | 15 (42.86%) | 8 (22.86%) | **0.01** |
| Attention Problems | 26 (74.29%) | 13 (37.14%) | **0.001** |
| Rule-Breaking Behavior | 11.43%)) 4 | 5.71%))2 | 0.208 |
| Aggressive Behavior | 17.14%)) 6 | 3 (8.57%) | 0.06 |
| Total score | 26 (74.29%) | 11 (31.43%) | **0.001** |

Data are expressed as number (%).

P: significance with active thyrotoxicosis versus in remission.

slow mental function) in 24% of subjects for a period of 2 years after the diagnosis of hyperthyroidism till the appearance of clinical symptoms of GD; (b) altered personality and mood or emotional functioning (manifested as depression, mood swings, anxiety and panic attacks) in 37%; and (c) somatic complaints (50%). The authors also observed that one third of patients received psychotropic medications after being diagnosed with GD and the majority had worse cognition and neuropsychiatric symptoms after achievement of peripheral normalized thyroid hormone levels compared to their baseline (i.e. before the onset of thyrotoxicosis). Scheffer et al. [5] studied females with the diagnosis of GD for five years and had normal thyroid hormone levels for ≤6 months before the study period. The authors observed the following: **(a)** high mean scoring values of Symptom Checklist-90-R (SCL-90-R), **(b)** psychological distress in 35.6% of patients [according to the Global Severity Index (GSI)], **(c)** higher scores in anxiety subscale [according to Hospital Anxiety and Depression Scale or HADS], and **(d)** higher relapse rates with higher scores of psychological distress (GSI more than 60). Vogel et al. [6] did a prospective study on adults (n = 31) who were newly diagnosed with GD and did not

**Table 4. Regression analysis for variables which predict behavioral problems.**

| | Unstandardized Coefficients | | Standardized Coefficients | t | Sig | 95% Confidence Interval for B | |
|---|---|---|---|---|---|---|---|
| | B | SE | β | | | Lower Bound | Upper Bound |
| **Children with active GD** | | | | | | | |
| Constant | 1.763 | 0.402 | - | 5.260 | **0.001** | 0.688 | 2.345 |
| TSH | 0.346 | 0.128 | 0.350 | 3.258 | **0.001** | 0.248 | 0.620 |
| fT4 | 0.460 | 0.205 | 0.352 | 3.520 | **0.001** | 0.246 | 0.845 |
| fT3 | 0.255 | 0.152 | 0.280 | 2.632 | **0.01** | 0.184 | 0.472 |
| TRAbs | 0.526 | 0.185 | 0.408 | 3.526 | **0.001** | 0.345 | 0.860 |
| **Children in normalized thyroid hormone levels** | | | | | | | |
| Constant | 1.624 | 0.326 | - | 4.346 | **0.001** | 0.456 | 2.530 |
| TSH | 0.322 | 0.138 | 0.320 | 2.356 | **0.03** | 0.268 | 0.562 |
| fT4 | 0.328 | 0.184 | 0.220 | 1.520 | **0.01** | 0.238 | 0.625 |
| fT3 | 0.246 | 0.128 | 0.152 | 1.824 | 0.06 | 0.145 | 0.526 |
| TRAbs | 0.433 | 0.256 | 0.368 | 3.228 | **0.001** | 0.265 | 0.625 |

TSH: thyroid stimulating hormone; fT4: free thyroxine; fT3: free triiodothyronine; TRAbs, thyroid-stimulating hormone receptor antibodies.

receive treatment yet. The authors observed the following: **(a)** higher scores of psychiatric rating scales, memory and concentration problems but not in neuropsychological test performances at baseline (untreated state), **(b)** no correlation between thyroid hormone levels and the neuropsychological test performances or psychiatric ratings at baseline; and **(c)** significant reduction of affective and somatic manifestations after reaching normal thyroid hormone levels and further normalization after ATD(s) treatment for one year, The authors concluded that the detected cognitive deficits might be the cause of the observed affective and somatic manifestations.

Furthermore, psychopathological and neuropsychological changes have been observed in adults with subclinical hyperthyroidism. Röckel et al. [18] observed anxiety, emotional irritability, depression, fatigability and lack of energy, poor attention and concentration in patients with subclinical hyperthyroidism similar to those with overt hyperthyroidism. Schlote et al. [19] observed the following: **(a)** increased frequency of anxiety symptoms and manifestations of hypermetabolic state in adults with subclinical hyperthyroidism (n = 35) compared to healthy subjects (n = 28), **(b)** similar manifestations of psychomotor impairment, hypermetabolic state and self-rating affective state in adults with subclinical hyperthyroidism as those with overt hyperthyroidism (n = 60), and **(c)** differences in the ability to concentrate and short-term memory among the three groups (i.e. subclinical hyperthyroidism, overt hyperthyroidism and healthy subjects).

The mechanisms underlying the development of psychiatric manifestations in GD and other causes of hyperthyroidism are not known. Studies indicated that psychological symptoms associated with hyperthyroidism are similar to those of primary anxiety and anxious depressive disorder (i.e. there are no specific or unique psychosomatic patterns in patients with hyperthyroidism). Therefore, psychological symptoms with hyperthyroidism might be related to: the biological consequence of autoimmunity, metabolic disorder of thyrotoxicosis and/or the associated neurotransmitter changes [17, 20, 21]. This is supported by the following: **(1)** the hippocampus and amygdala, brain regions involved in behavior, mood, and long-term memory have large number of thyroid hormone receptors [21]. **(2)** thyroid hormones have multiple actions, including: **(a)** modulation of noradrenergic (i.e. beta-adrenergic response to catecholamines in the central nervous system), serotonergic, and dopaminergic receptors' functions. **(b)** influencing effect on second messenger, calcium homeostasis and morphology of neuronal axons and their transport mechanisms. Animal studies reported that thyroxine increases serotonin (5HT2) receptors in the hippocampus and striatum [22]. **(3)** thyroid releasing hormone (TRH) is a neurotransmitter in the autonomic nervous system (ANS). It has also been found in peripheral lymphocytes [23]. Therefore, it is possible that hyperthyroidism may alter neurotransmitters' activities as monoamines, serotonin and dopamine in the limbic system resulting in neuropsychiatric manifestations [21]. This is also supported by the following: **(a)** Studies identified significant correlation between abnormal T4/T8 ratio and high scores of anxiety and depression in patients with GD [17]. **(b)** Studies observed that the duration and severity of thyrotoxicosis were the main determinants for behavioral complications [24]. **(4)** Studies observed that similar to patients with different systemic autoimmune diseases, psychiatric symptoms linked to thyrotoxicosis may occur earlier during the course of the disease or even before the diagnosis of integral hypermetabolic symptoms. In some adults, authors observed obvious neuropsychiatric symptoms antedating the clinical diagnosis of thyrotoxicosis by 6–12 months [17, 25, 26]. **(5)** studies found 45% increase in the risk for schizophrenia with the presence of any autoimmune disease [27]. On the other hand, higher prevalence rates of autoimmune diseases [e.g. autoimmune diseases included rheumatoid arthritis, celiac disease, autoimmune thyroid diseases, thyrotoxicosis, insulin dependent diabetes, acquired hemolytic anemia, interstitial cystitis, and Sjögren's syndrome] were reported in

patients with schizophrenia with crude incidence rate ratios ranging from 1.9 to 12.5. They also found adjusted incidence rate ratios of autoimmune diseases ranged from 1.3 to 3.8 in parents of patients with schizophrenia. **(6)** thyroxine and TRH are used successfully for treatment of major depression [28]. However, some recent studies and reviews showed controversial evidence [29]. **(7)** the occurrence of stress may play a role in evocation of psychiatric symptoms in genetically vulnerable patients with thyrotoxicosis [20, 28, 29]. In accordance, Morillo and Gardner [28] reported that the onset of relapse of GD in four children (8–14 years) (HLA B-8) was induced by depression as a common response to loss. The authors suggested that depression may result in the following: **(a)** suppression of immunity, **(b)** depletion of central nervous system monoamines, **(c)** activation of the hypothalamic-pituitary-adrenal (HPA) axis, and **(d)** formation of thyroid-stimulating immunoglobulins.

Therefore, it is necessary to: **(1)** identify the systemic autoimmune diseases earlier and treat them adequately, **(2)** increase the awareness of specialists with the neuropsychiatric complications of thyrotoxicosis, **(3)** recommend repeated psychological testing for children and adolescents with GD, and **(4)** individualize psychotropic therapy.

## The strengths of the study

The strengths of the study are **(1)** the pursuit of behavioral symptomatology in a previously unstudied population, and **(2)** The prospective nature of the study.

## Limitations of the study

We realize that the main limitation of this study is small sample size and the recruitment of patients from a single tertiary center may result in selection bias for severe cases. Future prospective studies with large sample size are recommended.

## Conclusions

This study indicates that children and adolescents with GD are at higher risk of behavioral symptoms even with normalized thyroid hormone levels after anti-thyroid drug therapy. Anxiety/Depression and inattention were the commonly associated behavioral symptoms with GD, indicating presence of externalizing and internalizing problems. The presence of behavioral changes during the active as well as normalized thyroid hormone levels' states (now euthyroid) highly suggests a biopsychological mechanism. Earlier diagnosis, optimizing management of GD and early screening for behavioral symptoms are important to improve patients' social life.

## Author Contributions

**Conceptualization:** Sherifa Ahmed Hamed, Fadia Ahmed Attiah, Samir Kamal Abdulhamid, Mohamed Fawzy.

**Data curation:** Sherifa Ahmed Hamed.

**Formal analysis:** Sherifa Ahmed Hamed.

**Investigation:** Sherifa Ahmed Hamed.

**Methodology:** Sherifa Ahmed Hamed, Fadia Ahmed Attiah, Samir Kamal Abdulhamid, Mohamed Fawzy.

**Writing – original draft:** Sherifa Ahmed Hamed, Fadia Ahmed Attiah, Samir Kamal Abdulhamid, Mohamed Fawzy.

**Writing – review & editing:** Sherifa Ahmed Hamed, Fadia Ahmed Attiah, Samir Kamal Abdulhamid, Mohamed Fawzy.

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
