## [Decision Letter · Decision Letter 0]

6 Jan 2021

PONE-D-20-38682

Behavioral assessment of children and adolescents with Graves' disease: A prospective study

PLOS ONE

Dear Dr. Hamed,

Thank you for submitting your manuscript to PLOS ONE. After careful consideration, we feel that it has merit but does not fully meet PLOS ONE’s publication criteria as it currently stands. Therefore, we invite you to submit a revised version of the manuscript that addresses the points raised during the review process.

We look forward to receiving your revised manuscript.

Kind regards,

Melissa A Brotman, PhD

Academic Editor

PLOS ONE

Additional Editor Comments:

Thank you very much for your manuscript to PLOS ONE. We have secured a review who saw great merits with the manuscript, but also raised some concerns, which are noted below. We encourage you to address the comments and submit a revised version. Thank you again for submitting to PLOS ONE.

Journal Requirements:

Reviewers' comments:

Reviewer's Responses to Questions

**Comments to the Author**

1. Is the manuscript technically sound, and do the data support the conclusions?

Reviewer #1: Yes

2. Has the statistical analysis been performed appropriately and rigorously? 

Reviewer #1: Yes

3. Have the authors made all data underlying the findings in their manuscript fully available?

Reviewer #1: Yes

4. Is the manuscript presented in an intelligible fashion and written in standard English?

Reviewer #1: No

5. Review Comments to the Author

Reviewer #1: 1. This report describes behavior in children with Graves disease prior to treatment with antithyroid drugs and following such treatment. Behavior is assessed using the Child Behavior Checklist.

2. The authors refer to the period during which the children with Graves disease are treated with anti-thyroid drugs as the 'remission state.' This description is misleading since 'remission' is commonly used to describe the phase of Graves disease in which treatment is not required as a result of decreased production of thyroid stimulating immunoglobulins or as a result of concurrent thyroiditis rendering the thyroid gland unable to respond to these stimulating immunoglobulins. A preferable term to describe normalization of thyroid hormone levels resulting from antithyroid drug therapy is 'normalized thyroid hormone levels.'

3. There are three occurrences in the manuscript in which the authors erroneously describe hyperthyroidism as 'hyperparathyroidism' :first and fifth sentences of the Introduction and third paragraph of the Discussion. Parathyroid hormone is produced by parathyroid glands (which are close to the thyroid gland. Parathyroid hormone regulates circulating levels of calcium and phosphorus.) This distinction is particularly important since parathyroid hormone excess can be associated with abnormal brain function.

4. Under 'Methods,' the second sentence should read: 'Their age ranged...'

5. Under 'Methods' in the first paragraph, third sentence: best to use the spelling 'thyroxine' rather than 'thyroxin'

6. Under 'Methods' in the first paragraph, fourth sentence: instead of 'thyroid hormone state' use 'thyroid hormone status.'

7. In the next sentence under 'Methods,' '(2) those in remission (ie, maintained in euthyroid state' best to spell euthyroid as above instead of 'euothyroid.' In addition best to refer to this state as normalized thyroid state' rather than as 'euthyroid.' The next sentence also has 'euthyroid' misspelled as 'euothyroid.'

8. Under 'Exclusion Criteria': the authors should consider including children with Graves ophthalmopathy since this condition should not confound findings in the WISC-III or in the CBCL/8-18.

9. Under 'Laboratory Testing': TSH should not be included under the term 'thyroid hormones.' Best to refer to all of these tests as 'thyroid function tests.'

10. Under 'Statistical analyses' eighth line--do not refer to status during treatment with antithyroid drugs as 'remission state.' Better to use 'drug-corrected state.'

11. In first paragraph under 'Results' and in first 2 paragraphs under 'Discussion': do not refer to drug-corrected state as 'remission.'

12. In second paragraph of the discussion, the authors should substitute 'normalized thyroid

thyroid hormone levels' for 'euthyroid state.'

13. In the first line on page 10 (part of the Discussion), the last word should be 'following' rather than 'followings.'

14. In second line on page 10: ' There are a large number...'

15. The authors refer to thyroid hormones as accepted treatment for depression. Recent studies and reviews question this. The authors should provide a recent reference supporting this practice.

6. PLOS authors have the option to publish the peer review history of their article (what does this mean?). If published, this will include your full peer review and any attached files.

Reviewer #1: **Yes: **Donald Zimmerman, MD

---

## [Author Response · Author response to Decision Letter 0]

15 Jan 2021

We are grateful to the reviewers for the valuable and helpful editor's and reviewers' comments on our manuscript ID # [PONE-S-20-48013] with the title "Behavioral assessment of children and adolescents with Graves' disease: A prospective study" submitted as an original article wishing to be published in Plos One. We have addressed all comments (point-by-point), as indicated in the attached pages and considered their corrections within the main text. We hope that the explanations and revisions of our work are satisfactory. We have highlighted the changes in the revised manuscript (yellow color). 

Response to reviewer's # 1 comments 

Reviewer:

1- This report describes behavior in children with Graves disease prior to treatment with antithyroid drugs and following such treatment. Behavior is assessed using the Child Behavior Checklist.

2- The authors refer to the period during which the children with Graves disease are treated with anti-thyroid drugs as the 'remission state.' This description is misleading since 'remission' is commonly used to describe the phase of Graves disease in which treatment is not required as a result of decreased production of thyroid stimulating immunoglobulins or as a result of concurrent thyroiditis rendering the thyroid gland unable to respond to these stimulating immunoglobulins. A preferable term to describe normalization of thyroid hormone levels resulting from antithyroid drug therapy is 'normalized thyroid hormone levels.'

Author:

We used the preferable term "normalized thyroid hormone levels state" instead of "in remission" whenever cited throughout the text. 

Reviewer:

3. There are three occurrences in the manuscript in which the authors erroneously describe hyperthyroidism as 'hyperparathyroidism': first and fifth sentences of the Introduction and third paragraph of the Discussion. Parathyroid hormone is produced by parathyroid glands (which are close to the thyroid gland. Parathyroid hormone regulates circulating levels of calcium and phosphorus.) This distinction is particularly important since parathyroid hormone excess can be associated with abnormal brain function.

Author:

They have been edited correctly

Reviewer:

4. Under 'Methods,' the second sentence should read: 'Their age ranged...'

Author:

It has been edited correctly

Reviewer:

5. Under 'Methods' in the first paragraph, third sentence: best to use the spelling 'thyroxine' rather than 'thyroxin'

Author:

We have replaced the term "thyroxin" with "thyroxine" throughout the manuscript text.

Reviewer:

6. Under 'Methods' in the first paragraph, fourth sentence: instead of 'thyroid hormone state' use 'thyroid hormone status.'

Author:

It has been edited.

Reviewer:

7. In the next sentence under 'Methods,' '(2) those in remission (ie, maintained in euthyroid state' best to spell euthyroid as above instead of 'euothyroid.' In addition best to refer to this state as normalized thyroid state' rather than as 'euthyroid.' The next sentence also has 'euthyroid' misspelled as 'euothyroid.'

Author:

We replaced euothyroid by "normalized thyroid state".

Reviewer:

8. Under 'Exclusion Criteria': the authors should consider including children with Graves ophthalmopathy since this condition should not confound findings in the WISC-III or in the CBCL/8-18.

Author: 

- None of the recruited children had Ophthalmopathy

- We revised the text of "previous relapse or presence of Graves Ophthalmopathy," into "previous relapse," 

Reviewer:

9. Under 'Laboratory Testing': TSH should not be included under the term 'thyroid hormones.' Best to refer to all of these tests as 'thyroid function tests.'

Author:

We have revised it as thyroid function tests

Reviewer:

10. Under 'Statistical analyses' eighth line--do not refer to status during treatment with antithyroid drugs as 'remission state.' Better to use 'drug-corrected state.'

Author:

We have revised it as 'drug-corrected

Reviewer:

11. In first paragraph under 'Results' and in first 2 paragraphs under 'Discussion': do not refer to drug-corrected state as 'remission.'

Author:

We have done and wherever found, we replace in remission by drug-corrected. 

Reviewer:

12. In second paragraph of the discussion, the authors should substitute 'normalized thyroid

thyroid hormone levels' for 'euthyroid state.'

Author:

We have substituted it by "normalized thyroid hormone levels"

Reviewer:

13. In the first line on page 10 (part of the Discussion), the last word should be 'following' rather than 'followings.'

Author:

It has been edited

Reviewer:

14. In second line on page 10: ' There are a large number...'

Author:

We have edited it. 

Reviewer:

15. The authors refer to thyroid hormones as accepted treatment for depression. Recent studies and reviews question this. The authors should provide a recent reference supporting this practice.

Author:

Page 10 and 11:

This has been revised as follow and we provided a recent reference supporting this practice as follow:

"(6) Thyroxine and TRH are used successfully for treatment of major depression [29,30]. However, some recent studies and reviews showed controversial evidence [31]."

Trifu S, Popescu A, Dragoi AM, Trifu AI. Thyroid hormones as a third line of augmentation medication in treatment-resistant depression. Acta Endocrinol (Buchar). 2020;16(2):256-261. http://doi.org./10.4183/aeb.2020.256. PMID: 33029246

Reviewer:

Any typographical or grammatical errors should be corrected at revision, so please note any specific errors here.

Author:

The text of manuscript has been revised by a colleague who naive language is English.

With this manuscript, there are 4 tables which we wish to be published.

I certify that the materials of this work “Behavioral assessment of children and adolescents with Graves' disease: A prospective study” have not been submitted simultaneously elsewhere and that there is no part of the text or any of the tables has been copyrighted, published, or reproduced elsewhere. I certify that all individuals included as authors of papers have contributed substantially to the scientific process leading up to the writing of the paper and its revised version. 

Thank you for accepting reviewing our manuscript.

---

## [Decision Letter · Decision Letter 1]

10 Feb 2021

PONE-D-20-38682R1

Behavioral assessment of children and adolescents with Graves' disease: A prospective study

PLOS ONE

Dear Dr. Hamed,

Thank you for submitting your manuscript to PLOS ONE. After careful consideration, we feel that it has merit but does not fully meet PLOS ONE’s publication criteria as it currently stands. Therefore, we invite you to submit a revised version of the manuscript that addresses the points raised during the review process.

We look forward to receiving your revised manuscript.

Kind regards,

Melissa A Brotman, PhD

Academic Editor

PLOS ONE

Reviewers' comments:

Reviewer's Responses to Questions

**Comments to the Author**

1. If the authors have adequately addressed your comments raised in a previous round of review and you feel that this manuscript is now acceptable for publication, you may indicate that here to bypass the “Comments to the Author” section, enter your conflict of interest statement in the “Confidential to Editor” section, and submit your "Accept" recommendation.

Reviewer #1: (No Response)

2. Is the manuscript technically sound, and do the data support the conclusions?

Reviewer #1: Yes

3. Has the statistical analysis been performed appropriately and rigorously? 

Reviewer #1: Yes

4. Have the authors made all data underlying the findings in their manuscript fully available?

Reviewer #1: Yes

5. Is the manuscript presented in an intelligible fashion and written in standard English?

Reviewer #1: No

6. Review Comments to the Author

Reviewer #1: 1. Abstract-1st sentence: Previous studies have...instead of

2. Abstract-second sentence: These studies are scarce [not scare]

3. Abstract -3rd sentence: This study aimed to STUDY [rather than 'determine.']

4. Abstract- The authors appear to associate worse outcomes with low TSH levels and then with higher TSH levels (both in abstract and in first paragraph of Results. Please explain these apparently contradictory findings.

7. PLOS authors have the option to publish the peer review history of their article (what does this mean?). If published, this will include your full peer review and any attached files.

Reviewer #1: **Yes: **Donald Zimmerman, MD

---

## [Author Response · Author response to Decision Letter 1]

12 Feb 2021

We are grateful to the reviewers for the valuable and helpful editor's and reviewers' comments on our manuscript ID # [PONE-D-20-38682R1] with the title "Behavioral assessment of children and adolescents with Graves' disease: A prospective study" submitted as an original article wishing to be published in Plos One. We have addressed all comments (point-by-point), as indicated in the attached pages and considered their corrections within the main text. We hope that the explanations and revisions of our work are satisfactory. We have highlighted the changes in the revised manuscript (yellow color). 

Response to reviewer's # 1 comments 

Reviewer:

1. Abstract-1st sentence: Previous studies have...instead of

Author:

It has been revised to "Previous studies have identified frequent…."

Reviewer:

2. Abstract-second sentence: These studies are scarce [not scare]

Author:

It has been corrected.

Reviewer:

3. Abstract -3rd sentence: This study aimed to STUDY [rather than 'determine.']

Author:

It has been revised as follow:

"This work aimed to study…."

Reviewer:

4. Abstract- The authors appear to associate worse outcomes with low TSH levels and then with higher TSH levels (both in abstract and in first paragraph of Results. Please explain these apparently contradictory findings.

Author:

This has been clarified as in the abstract as the results of the study as follow: 

Abstract:

"Multiple linear regression analysis showed that lower concentrations of TSH (P=0.001; P=0.03) and higher fT4 (P=0.001, P=0.01), fT3 (P=0.01; P=0.06) and TRAbs (P=0.001; P=0.001) (at presentation)"

Results:

"Multiple linear regression analysis showed that "at presentation" lower concentrations of TSH (P=0.001; P=0.03) and higher fT4 (P=0.001, P=0.01), fT3 (P=0.01; P=0.06) and TRAbs (P=0.001; P=0.001) were predictors of behavioral problems in children with active GD and normalized thyroid hormone levels' states (table 4)."

Reviewer:

Any typographical or grammatical errors should be corrected at revision, so please note any specific errors here.

Author:

The text of manuscript has been revised by a colleague who naive language is English.

With this manuscript, there are 4 tables which we wish to be published.

I certify that the materials of this work “Behavioral assessment of children and adolescents with Graves' disease: A prospective study” have not been submitted simultaneously elsewhere and that there is no part of the text or any of the tables has been copyrighted, published, or reproduced elsewhere. I certify that all individuals included as authors of papers have contributed substantially to the scientific process leading up to the writing of the paper and its revised version. 

Thank you for accepting reviewing our manuscript.

---

## [Decision Letter · Decision Letter 2]

3 Mar 2021

PONE-D-20-38682R2

Behavioral assessment of children and adolescents with Graves' disease: A prospective study

PLOS ONE

Dear Dr. Hamed,

Thank you for submitting your manuscript to PLOS ONE. After careful consideration, we feel that it has merit but does not fully meet PLOS ONE’s publication criteria as it currently stands. Therefore, we invite you to submit a revised version of the manuscript that addresses the points raised during the review process.

We look forward to receiving your revised manuscript.

Kind regards,

Melissa A Brotman, PhD

Academic Editor

PLOS ONE

Reviewers' comments:

Reviewer's Responses to Questions

**Comments to the Author**

1. If the authors have adequately addressed your comments raised in a previous round of review and you feel that this manuscript is now acceptable for publication, you may indicate that here to bypass the “Comments to the Author” section, enter your conflict of interest statement in the “Confidential to Editor” section, and submit your "Accept" recommendation.

Reviewer #1: All comments have been addressed

2. Is the manuscript technically sound, and do the data support the conclusions?

Reviewer #1: Yes

3. Has the statistical analysis been performed appropriately and rigorously? 

Reviewer #1: Yes

4. Have the authors made all data underlying the findings in their manuscript fully available?

Reviewer #1: Yes

5. Is the manuscript presented in an intelligible fashion and written in standard English?

Reviewer #1: Yes

6. Review Comments to the Author

Reviewer #1: 1. The manuscript is improved. Most of the problems which remain are related to language rather than to scientific issues.

2. Abstract Line 7: Instead of 'active and normalized thyroid hormone levels' states'---'during periods of thyroid hormone elevation and of normalized thyroid hormones.' Use these terms also in line 16. Similar verbiage should be used in line 4 and in the final paragraph of the introduction.

3.In 'Methods' section--line 3: omit comma between 'out-patient clinic' and and 'of Assiut University Hospital.'

4. 'Methods' section line 8:

(2) those with normal thyroid hormone levels

5. 'Methods' section line 10:

'...which was subsequently titrated to maintain normal thyroid hormone levels based on...'

7. PLOS authors have the option to publish the peer review history of their article (what does this mean?). If published, this will include your full peer review and any attached files.

Reviewer #1: **Yes: **Donald Zimmerman, MD

---

## [Author Response · Author response to Decision Letter 2]

4 Mar 2021

We are grateful to the valuable and helpful reviewers' and editor's comments on our manuscript ID # [PONE-D-20-38682R2] with the title "Behavioral assessment of children and adolescents with Graves' disease: A prospective study" submitted as an original article wishing to be published in Plos One. We have addressed all comments (point-by-point), as indicated in the attached pages and considered their corrections within the main text. We hope that the explanations and revisions of our work are satisfactory. We have highlighted the changes in the revised manuscript (yellow color). 

Response to Editor's comments 

- The text of manuscript has been edited by a colleague who naive language is English.

- We have included this statement in your website related to Data Availability "Data cannot be shared publicly because they contain potentially sensitive information. Data are available from the ethics committee of Faculty of Medicine, Assiut, University (medicinegraduate@aun.edu.eg)".

Response to reviewer's # 1 comments 

Reviewer:

1. The manuscript is improved. Most of the problems which remain are related to language rather than to scientific issues.

Author:

Reviewer:

2. Abstract Line 7: Instead of 'active and normalized thyroid hormone levels' states'---'during periods of thyroid hormone elevation and of normalized thyroid hormones.' Use these terms also in line 16. Similar verbiage should be used in line 4 and in the final paragraph of the introduction.

Author:

We have done the required corrections

Reviewer:

3. In 'Methods' section--line 3: omit comma between 'out-patient clinic' and and 'of Assiut University Hospital.'

Author:

It has been corrected.

Reviewer:

4. 'Methods' section line 8: (2) those with normal thyroid hormone levels

Author:

It has been corrected.

Reviewer:

5. 'Methods' section line 10: '...which was subsequently titrated to maintain normal thyroid hormone levels based on...'

Author:

It has been corrected.

With this manuscript, there are 4 tables which we wish to be published.

I certify that the materials of this work “Behavioral assessment of children and adolescents with Graves' disease: A prospective study” have not been submitted simultaneously elsewhere and that there is no part of the text or any of the tables has been copyrighted, published, or reproduced elsewhere. I certify that all individuals included as authors of papers have contributed substantially to the scientific process leading up to the writing of the paper and its revised version. 

Thank you for accepting reviewing our manuscript.

---

## [Editor Report · Decision Letter 3]

9 Mar 2021

Behavioral assessment of children and adolescents with Graves' disease: A prospective study

PONE-D-20-38682R3

Dear Dr. Hamed,

We’re pleased to inform you that your manuscript has been judged scientifically suitable for publication and will be formally accepted for publication once it meets all outstanding technical requirements.

Kind regards,

Melissa A Brotman, PhD

Academic Editor

PLOS ONE
---

## [Editor Report · Acceptance letter]

11 Mar 2021

PONE-D-20-38682R3 

Behavioral assessment of children and adolescents with Graves' disease: A prospective study 

Dear Dr. Hamed:

I'm pleased to inform you that your manuscript has been deemed suitable for publication in PLOS ONE. Congratulations! Your manuscript is now with our production department. 

Kind regards, 

on behalf of

Dr. Melissa A Brotman 

Academic Editor

PLOS ONE